



# The solar induced 27-day modulation on polar mesospheric cloud (PMC), based on the combined observations from SOFIE and MLS

Shican Qiu[1], Mengzhen Yuan[1,2], Willie Soon[3,4], Victor Manuel Velasco Herrera[5], Zhanming Zhang[1,2], Xiankang Dou[2*]

[1] Department of Geophysics, College of the Geology Engineering and Geomatics, Chang'an University, Xi'an, 710054, China

[2] Key Laboratory of Geospace Environment, Chinese Academy of Sciences, University of Science & Technology of China, Hefei, Anhui, 230026, China

[3] Center for Environmental Research and Earth Sciences (CERES), Salem, MA 01970, USA

[4] Institute of Earth Physics and Space Science (ELKH EPSS), 9400, Sopron, Hungary

[5] Instituto De Geofísica, Universidad Nacional Autónoma De México, Mexico City, Mexico

*Correspondence to*: Shican Qiu (scq@ustc.edu.cn ) Xiankang Dou (dou@ustc.edu.cn)

**Abstract.** Temperature is considered to be the key driving factor of the polar mesospheric cloud (PMC) variations, and the external source of temperature change is mainly from the solar radiations. In this paper, we use the observations of vertical column of ice water content (IWC) and mesopause temperature collected by the Solar Occultation For Ice Experiment (SOFIE), combined with the temperature data of Microwave Limb Sounder (MLS), to determine the time lags between temperature and IWC anomalies in responding to the solar radiation index Y10, through superposed epoch analysis (SEA) and time lag correlation analysis methods. The results show that the IWC responses to the Y10 later than the mesospheric temperature does. Further investigation of the relationship between mesospheric temperature and PMC reveals that the average time lag day is 0 days in the northern hemisphere (NH), and 1 day in the southern hemisphere (SH). The differences in temperature response to the 27-day solar rotational modulation with atmospheric pressure and latitude are analyzed, based on the temperature observations from 2004 to 2020 by MLS. Twelve PMC seasons with 27-day periodicity are distinguished, with 9 of them have time lags increasing with atmospheric pressure (or decreasing with altitude).

**Key words**: polar mesospheric clouds (PMCs), solar radiation, Y10, ice water content (IWC), temperature



## 1 Introduction

The polar mesospheric cloud (PMC) is formed in the mesopause region (80–90 km) over the high latitude (> 50°) during summer times (Hervig et al, 2012). The PMC is mainly composed of small water ice crystals (with radius about 30–100 nm), which can effectively scatter sunlight to make it visible before dusk and dawn (Hervig et al, 2012; Dalin et al., 2018). The

extremely low temperature (T<140 K) at the summer mesopause, accompanied by an increment of water vapor content, will result in a supersaturated state for the formation of water ice aerosols (Hervig et al, 2001). In the Northern Hemisphere (NH), a typical PMC season lasts from late May to the end of August (Gadsden, 1982; Gadsden, 1998), while appears from late November to mid-February in the Southern Hemisphere (SH) (Ludlam, 1976; Thomas, 1985).

The PMCs are very sensitive to changes in the mesopause environment and are influenced by many atmospheric parameters,

such as the temperature (Thomas et al., 2015), water vapor content (Thomas et al., 2015), gravity waves (Gerrard et al., 2004; Chandran et al., 2010), planetary waves (Merkel et al., 2003; von Savigny et al., 2007), and the interhemispheric coupling (Karlsson et al., 2007; Karlsson et al., 2009). Therefore, the PMCs are considered as an important indicator of the mesospheric state.

The solar activity and its variations can affect PMCs by affecting the temperature and water vapor in the summer

mesopause (Hervig et al., 2006). Due to the sun's axial rotation and the unique spatial distributions of magnetic features on the surface of the Sun like sunspots and other brighter components like plages and faculae, the solar radiation received by the earth is quasi–periodic, with the main period of 27 days (e.g., roughly the Carrington solar rotation period), and weaker periodic modulation of 13 days and 9 days or so (Lean et al., 1987). This is why and how the 27-day periodic modulation and disturbance of the light and charged particle outputs from the Sun will affect the radiative photochemical and kinetic state of the mesosphere

(Hood et al., 1991; Beig et al., 2008).

Observations from satellites and ground-based equipments show that the frequency of occurrence (Robert et al., 2010; von Savigny et al., 2013), albedo (von Savigny et al., 2013), ice water content (Thurairajah et al., 2017) and brightness (Dalin et al., 2018) of the PMCs are characterized by a 27-day periodicity. This period has been proposed to be physically connected with the solar $Lyman-\alpha$ flux with a time lag of 0 to 1 day (Robert et al., 2010; von Savigny et al., 2013; Thurairajah et al.,

2017; Dalin et al., 2018). Since the solar radiation will increase the mesopause temperature by heating, the temperature may remain above the freezing point with large enough heat from the Sun (Robert et al., 2010; von Savigny et al., 2013). And the solar radiation can photolyze water vapor, further weakens or prevents the PMCs and its formation (Robert et al., 2010; von Savigny et al., 2013). Simulation of the of 27-day solar ultraviolet forcing effect on the mesospheric temperature shows the 27-day solar cycle is intermittent and may further depends on atmospheric dynamics (Gruzdev et al., 2009).

The superposed epoch analysis (SEA) can extract the signals of the PMCs in response to the solar forcing from the background noise, and the data can be properly averaged to amplify the useful signals by eliminating the noise (e.g. Robert et al., 2010; Thomas et al., 2015). Using this method, the analysis of temperature observation from the Solar Occultation For Ice Experiment (SOFIE) onboard the Aeronomy of Ice in the Mesosphere (AIM) exhibits a positive and faster response to solar



*Lyman−α* than due to the direct solar heating, suggesting an effect on the 27-day vertical wind variability (Thomas et al.,
2015). The average time lag days (e.g., with 0–3days) of long-term PMC brightness to the solar *Lyman−α* flux is shorter,
indicating that direct solar heating is the main dynamic mechanism, and the solar *Lyman−α* flux plays a secondary role in the
photolysis of water vapor (Dalin et al., 2018).

        In this research, the IWC and temperature measured by SOFIE and additional mesospheric temperature observations from
the Microwave Limb Sounder (MLS) onboard Aura satellite are co-analyzed. The solar activity index, Y10, includes both $X -$
$ray$ and $Lyman - α$ (Tobiska, 2010). Combined with the Y10, the SEA is used to extract the time lag of IWC and temperature
in responding to solar forcing over NH and SH, respectively. In addition, the cross-correlation curves between the average
temperature and solar forcing for different atmospheric pressure and latitudes of the PMC seasons from 2004 to 2020 are also
compared and analyzed. The data processing methods are presented in Section 2. Section 3 introduces the data used in this
study. Section 4 gives the results and explanations. Finally, Section 5 summarizes the study.

**2 Data and Method**

**2.1 The measurements and data pre–processing**

The Solar Occultation For Ice Experiment (SOFIE) is one of three scientific instruments onboard the Aeronomy of Ice in the
Mesosphere (AIM) satellite, which is dedicated to study the PMCs and their formation environment for the first time (Gordley
et al., 2009). The detailed description of the SOFIE instruments can be found in Gordley et al. (2009). It measures the reduction
of solar intensity when light passes through atmospheric tangent path at sunset or sunrise from orbit, based on the solar
occultation method. SOFIE provides 15 sunrise measurements from ~60°–85°S and 15 sunset measurements from ~60°–85°N
per day, essentially after the day-night terminator. We use the version 1.3 temperature and IWC data available from SOFIE
website ([http://gats-inc.com](http://gats-inc.com)).

        The Microwave Limb Sounder (MLS), carried on the Aura satellite and launched on July 15, 2004, measures the thermal
microwaves of atmospheric substances by five spectra from 115 GHz to 2.5 THz through limb observation (Waters et al.,
2006). The MLS scans vertically with spatial coverage of nearly global scale (with latitude of −82° to +82°) (Waters et al.,
2006). Each profile is 1.5° or 165 km along the orbit (e.g., approximately 15 orbits per day), with vertical coverage ranging
from 0.0010 pa ~100,000 pa (Schwartz et al., 2008). The vertical scan rate varies with altitude and is faster in the upper
atmosphere, resulting in poor vertical resolution in the mesosphere (Robert et al., 2010). In this paper, we use the version 5.0
temperature data from the [Microwave Limb Sounder (MLS) | Earthdata (nasa.gov).](https://nasa.gov)

        In order to analyze the correlation, the seasonal components in the time series need to be eliminated. The variation of
solar radiation intensity causes the fluctuation of temperature and IWC upon the background values. According to the previous
studies, 35-day window can be used for the smoothing filter, which contains an entire 27-day period (Hood et al., 1991; Robert
et al., 2010; Thomas et al., 2015). Then the time series of the anomalies can be obtained by subtracting the running mean of
35 days from the original time series (e.g.Hood et al., 1991; Robert et al., 2010; Thomas et al, 2015). Without the seasonal



components, we further use a 5-day running mean to smooth the anomalous time series. Since the smoothing process would cause the same phase shift to all three datasets, temperature, IWC, and Y10, the effect of smoothing on the results of the following correlation analysis is almost negligible.

An example of the smoothing of Y10 during the 2016 PMC season of NH is shown in Figure 1(a). And Figure 1(b) exhibits the average temperature in the latitude range of $60°–80°$ at 0.10 pa atmospheric pressure level. Figure 1(c) shows the curve of Y10 anomaly and temperature anomaly with relative solstice days, and there is a delayed change in the temperature anomaly. Figure 1(d) shows the correlation coefficient of Y10 anomaly and temperature anomaly in different time lag days. The time lag days corresponding to the two peaks of the correlation curve are $–19$ days and 8 days, respectively, with the absolute value in total equaling to $|-19| + 8 = 27$. The periodic variations reveal that there are 27-day characteristics of solar cycle in the mesospheric temperature dataset, which is consistent with previous results.

### 2.2 The cross-correlation analysis and significance test

The Spearman rank correlation coefficient is stable and insensitive to noise. Any two variables $x$ and $y$ will be sorted according to their sizes, with the sample number of $n$. The sorted serial numbers are denoted as $D_x^i$ and $D_y^i$, respectively. Then the Pearson correlation coefficient between the new serial numbers is the Spearman rank correlation coefficient ($r_s$) (Spearman, 2010), given as:

$$r_s(x,y) = \frac{\sum_{i=1}^{n}(D_x^i - \overline{D_x})(D_y^i - \overline{D_y})}{\sqrt{\sum_{i=1}^{n}(D_x^i - \overline{D_x})^2 \sum_{i=1}^{n}(D_y^i - \overline{D_y})^2}}, \tag{1}$$

and simplified to

$$r_s(x,y) = 1 - \frac{6\sum_{i=1}^{n}(D_x^i - D_y^i)^2}{n(n^2-1)}. \tag{2}$$

In one PMC season, the selected time series of mesospheric temperature anomalies is:

$$T\_anomalies(day_1, \dots \dots, day_n). \tag{3}$$

When the delay–day parameter is set as lags, the corresponding time series of Y10 anomalies will be:

$$Y10\_anomalies(day_1 - lags, \dots \dots, day_n - lags). \tag{4}$$

Then their rank correlation coefficient is $r_s$ ($T\_anomalies, Y10\_anomalies$), for the time lag correlation analysis.

The significance of the rank correlation coefficient can be tested using the Z statistic, which is calculated as:

$$Z = \frac{r_s}{\sqrt{1/(n-1)}}. \tag{5}$$

If a significant level of 0.05 is given, $Z$ will equal to 1.65, according to the Table A1 from the Appendix. Then $r_s = 0.21$ is calculated with data length $n = 61$. If the significant level is 0.01, then $Z = 2.33$ from the Table A1. When the data length





$n = 61$, $r_s = 0.30$ is calculated. Therefore, when the calculated rank correlation coefficient is higher than 0.21, the significance level is more than 95%. When the calculated rank correlation coefficient is higher than 0.30, the significance level can be more than 99%.

### 2.3 The superposed epoch analysis (SEA)

The superposed epoch analysis (SEA) can extract the response signal under certain forcing from the random noise (Chree, 1913), and this method is widely used to study the response of PMCs to the 27-day solar forcing (e.g.Robert et al., 2010; Thomas et al., 2015). We set the time when the maximum value of Y10 anomaly is located as the key date, which is designated as day 0. Then a 35-day time series of Y10 anomaly with obvious 27-day period is selected from the PMC seasons. The Y10 anomalies of different seasons are arranged together by rows and then averaged by columns to obtain a single 35-day Y10 anomaly time series (super-forcing function). Similarly, doing the same process for the temperature and IWC on the corresponding dates will yield two super-response functions. By averaging, the noise in the data will be much reduced because random noise cancels each other out. Table 1 shows the PMC seasons and the key dates of the selected PMC seasons used for the SEA in this paper. Some seasons have no obvious 27-day period of Y10 anomalies, so those seasons were excluded for the SEA (marked with asterisks in Table 1).

### 3 Data Processing

### 3.1 Solar activity index and PMCs indicator

The Y10, including both $X-ray$ and $Lyman-\alpha$, is selected as the index of solar activity (Tobiska, 2010). The $X-ray$ in the wavelength ranges between 0.1–0.8 nm comes from the cold and hot corona, usually combined with very bright and slowly changing (with time scale of day to month) solar activity region and rapidly varying (minute to hour) flares (Tobiska et al., 2008). Its photons reaching the Earth are absorbed by molecular oxygen ($O_2$) and molecular nitrogen ($N_2$) at the upper mesosphere and lower thermosphere (MLT, e.g., 85–100 km) (Nicolet et al., 1960). They can ionize the neutral components, and create the D region of the ionosphere (Thomson et al., 2004). The $Lyman-\alpha$ is produced in the upper chromosphere and transition region of the Sun (Woods et al., 1997). It is mainly formed in the solar active area and the flocculus (Woods et al., 1997). It will also be absorbed in the MLT region, decomposing $NO$ (Woods et al., 1997) and participating in the chemical reaction of $H_2O$ (Frederick, 1977). The Y10 is obtained by weighting the $X-ray$ and $Lyman-\alpha$, without a flare component (Tobiska, 2010). During high solar activity, the main energy source of the MLT is the $X-ray$ radiation, while during moderate and low solar activity it is the $Lyman-\alpha$ radiation (Tobiska, 2010). For the intermediate phase between maximum and minimum of the 11-year-like sunspot and solar activity cycles, the $X-ray$ and $Lyman-\alpha$ are competitive in the MLT region.





Then we will investigate the influence of solar radiation on PMCs in two steps. The first step is to study the response of mesopause temperature and IWC to the 27-day periodic variation of Y10. The second step studies the response of IWC to temperature variation, linking the effect of solar radiation on PMCs through temperature to IWC. As mentioned above, PMCs are affected by different atmospheric physical processes, many of which are not directly related to solar radiation. Therefore, appropriate parameters must be filtered out as indicators of PMC properties. Since the PMCs are mostly composed of small ice crystals, the IWC data collected by SOFIE in 9 NH PMC seasons (from 2007 to 2015) and 8 SH PMC seasons (from 2007/2008 to 2014/2015) are processed as the index. The variations of IWC can directly reflect the nature of the PMCs. The data of 30 days before and after the summer solstice are selected for each PMC season, when the supersaturation occurs.

### 3.2 The temperature data over the mesopause

The temperaute is considered to be the main driving factor of the 27-day periodic variation of PMCs (Robert et al., 2010; von Savigny et al., 2013). Microphysical model results reveal that the mass density of PMCs is weakly dependent on the abundance of condensation nuclei (Megner, 2011). According to the previous study, ice particles will condense and grow when the saturation of water vapor ($Sa$) is greater than 1 (Thomas, 1996). The calculation of $Sa$ is given as follows:

$$Sa = \frac{w(H_2O) \times p}{e^{(28.548-(6077.4 \div T))}} = \frac{e^{(6077.4 \div T)}}{e^{(28.548)}} \times w(H_2O) \times p \ , \tag{6}$$

where $p$ is the total gas pressure in $Nm^{-2}$, and $w(H_2O)$ is the water vapor mixing ratio (Thomas, 1996).

From the Eq. (6), we can see that $Sa$ is exponentially related to temperature but linearly related to water vapor. Therefore, in order to simplify the relationship between PMCs and solar activity, the effect of water vapor can be ignored and only the temperature is chosen as the intermediate variable. We selected data at three atmospheric pressures of 0.10 pa, 0.22 pa, and 0.46 pa, within the latitude range of 81° in the NH and SH in summer for the period 2004–2020 for analyzing the spatial and temporal variations of temperature response to solar forcing. Then we select data at three atmospheric pressures at 0.10 pa, 0.22 pa, and 0.46 pa, within the latitude range of $53° \leq |\emptyset| \ll 81°$ in the NH and SH from 2004 to 2020, for studying the dependence on latitudes.

### 4 Results and discussions

### 4.1 Analysis of SEA results

It is assumed that the response follows the peak forcing and the time lag is always positive (Thomas et al., 2015). According to previous results, solar forcing is positively correlated with temperature (Thomas et al., 2015; Köhnke et al., 2018) and negatively correlated with IWC (Thurairajah et al., 2017). Therefore, it can be stipulated that the positive delay days corresponding to the maximum correlation coefficient are the time lags of temperature response, and the positive delay days corresponding to the minimum correlation coefficient are the time lags of IWC response. Therefore, the days of positive delay





corresponding to the maximum correlation can be specified as the time lags of temperature response, and the positive delay days corresponding to the minimum correlation coefficient are the time lags of IWC.

Figure 2 shows the average response of SOFIE and MLS temperatures to the solar forcing for both hemispheres. The results reveal that the time lags with statistical significance >95% of the NH temperature response to Y10 are 2–5 days (based on data from SOFIE) and 0–7 days (for MLS dataset), respectively. While in SH, the time lag days are 5–17 days (for SOFIE) and 7–18 days (for MLS), respectively. With significant differences, the time lags in SH are much longer than those in NH. Figure 3 exhibits the average response of the IWC to solar forcing for the NH (left) and SH (right). The time lag days in the NH are 0–8 days (with statistically significant > 95 % of all days), and the time lag days in the SH are 14 days. The trough in the NH is lower than that in the SH, indicating that statistical visibility is higher in NH. Previous results show that the IWC anomalies are negatively correlated with $Lyman-\alpha$, with 0–5 days in NH and 3–8 days in SH (Thurairajah et al., 2017). In contrast, our results are closer to those in the NH but larger in the SH. The SEA reveals the relationship between IWC and Y10, that is there exists a delayed response between PMC and solar activity, with longer time lag days in the SH than in the NH.

The average response of IWC to temperature in NH versus SH is shown as Figure 4. The delay days corresponding to the trough are the time lags of the response of IWC to the temperature anomalies. The trough below the red line indicates that the statistical significance is > 95 %. The time lag in the NH is always 0 day, shown as Figure 4(a) and 4(c). In the SH, time lag is 1 day (shown as Figure 4(d), based on data from MLS statistically significant > 95 %) and −1 day (shown as Figure 4(b), based on SOFIE with statistical insignificance). It indicates that when the temperature changes in the NH, the IWC can respond rapidly, and the average time lag is 0 days. Therefore, the temperature anomaly in NH can directly reflect the PMC activity. Temperature associates solar forcing with PMC, through the impacts of temperature variations on the properties of PMC synchronously. In contrast, the IWC in SH is less sensitive to temperature anomaly, with time lag of 1 day.

### 4.2 Further study on the effect of Y10 on the mesospheric temperature

### 4.2.1 The cases having 27-day period with time lags monotonically decreasing with latitude

Figure 5 shows the average response of temperature anomaly and Y10 anomaly for the three PMC seasons of NH 2005 (a), NH 2009 (b), and SH 2005/2006 (c) over different latitudes at 0.1 pa atmosphere pressure. It can be seen from the figure that there is a significant phase shift of the correlation curve from low latitude to high latitude in the negative direction. This result is consistent with previous study, e.g., Gruzdev et al. (2009) have found that the sensitivity of mesospheric temperature to solar activity usually decreases with the latitude when using a three-dimensional chemical climate model. A possible explanation is that in the case of uniform distribution of incoming solar irradiance, the high latitudes are more sensitive to solar activity with faster response, and the low latitude area has weak sensitivity to solar activity with longer response time.



### 4.2.2 The periodic cases having 27-day period with time lags increasing with latitude or almost unchanged

Figure 6 reveals the average response of temperature anomaly and Y10 anomaly for the four PMC seasons of NH 2012 (a), NH 2018 (b), SH 2013/2014 (c), and SH 2017/2018 (d) over different latitudes at 0.1 pa atmospheric pressure. The result exhibits a significant phase shift of the correlation curve from low latitude to high latitude in the positive direction. Figure 7

shows the time lag in the PMC seasons of NH 2016 (a), NH 2019 (b), SH 2008/2009 (c), SH 2012/2013 (d), SH 2014/2015 (e) and SH 2015/2016 (f) is almost constant. The possible explanation is that the annually varying atmospheric dynamical state changes the sensitivity of temperature to solar activity, resulting in interannual variations in the time lag days of the mesosphere temperature response (Gruzdev et al., 2009). Moreover, the correlation curves at distinct latitudes also reflects the differences in the distribution of solar radiation, with the contribution of solar forcing to temperature variability increasing where solar

radiation is strong. Where solar radiation is weak, the influence of solar forcing on PMC is easily concealed or hidden by other atmospheric physical processes, possibly resulting in a weaker correlation between temperature and solar radiation. The time lags of temperature response do not change significantly with latitude, which may be caused by the uniform distribution of solar radiation index in latitude. The unchanged time lag with latitude may be caused by the uniform distribution of the solar radiation over latitudes.

### 4.2.3 Comparison of time lag days at different atmospheric pressure levels

We select 12 PMC seasons with obvious 27-day periodicity detected. The average time lags in the latitude range of 53°–81° at three different atmospheric pressure (e.g., 0.10 pa, 0.22 pa and 0.46 pa) are compared, and the statistical results are summarized in Table 2. Among them, 9/12 cases have a distinct pattern of increasing time lags with increasing atmospheric pressure (decreasing altitude), that is, the lower the altitude, the longer the time lags. These cases are marked in bold in Table

2, which are PMC seasons of NH 2009, NH 2016, NH 2019, SH 2005/2006, SH 2008/2009, SH 2013/2014, SH 2014/2015, SH 2015/2016, and SH 2017/2018. The possible explanation is that since solar radiation contacts the upper atmosphere first, it will take the lead in responding. The difference in response time of different PMC seasons at different altitude may be related to the varying timescale of atmospheric downward heat conduction caused by solar radiation.

### 5 Conclusions

In this paper, the average response of IWC to temperature in the PMC seasons of both hemispheres is obtained by using the SEA. The correlation of the average temperature and Y10 anomalies at different latitudes at 0.10 Pa atmospheric pressure in 15 NH and 16 SH PMC seasons from 2004 to 2020 are divided into four categories. The time lag (in days) of temperature at different altitude in with obvious 27-day periodicity are compared. The main conclusions of this paper are as follows:

1. Based on the SEA and time lag correlation analyses, it is found that when the temperature changes abnormally in

responding to the solar rotational modulation on the 27-day timescale, the IWC can respond rapidly, with an average response time lag of 0 day in the NH and 1 day in the SH.





2. According to the cross-correlation curves between the temperature anomaly and Y10 at the atmospheric pressure of 0.1 pa, the PMCs season can be possibly divided into four categories: i) with obvious 27-day period, the time lag of temperature monotonically decreases with latitude, ii) the time lag monotonically increases with latitude, iii) the time lag almost remains

unchanged with latitude, iv) with no obvious 27-day period.

3. Comparing the average response to 27-day solar forcing at latitudes of 53°–81° at 0.10 pa, 0.22 pa and 0.46 pa in different PMCs seasons, it is found that the time lag increases with the atmospheric pressure (i.e.., decreases with altitude).

**Appendices**

Table A1. Cumulative standardized normal distribution function (Ross 2014)

$F(z) = \frac{1}{\sqrt{2\pi}} \int_{-\infty}^{z} e^{-\frac{z^2}{2}} dz,$

| z | .00 | .01 | .02 | .03 | .04 | .05 | .06 | .07 | .08 | .09 |
|------|--------|--------|--------|--------|--------|--------|--------|--------|--------|--------|
| 0.50 | 0.6915 | 0.6950 | 0.6985 | 0.7019 | 0.7054 | 0.7088 | 0.7123 | 0.7157 | 0.7190 | 0.7224 |
| 0.60 | 0.7257 | 0.7291 | 0.7324 | 0.7357 | 0.7389 | 0.7422 | 0.7454 | 0.7486 | 0.7517 | 0.7549 |
| 0.70 | 0.7580 | 0.7611 | 0.7642 | 0.7673 | 0.7704 | 0.7734 | 0.7764 | 0.7794 | 0.7823 | 0.7852 |
| 0.80 | 0.7881 | 0.7910 | 0.7939 | 0.7967 | 0.7995 | 0.8023 | 0.8051 | 0.8078 | 0.8106 | 0.8133 |
| 0.90 | 0.8159 | 0.8186 | 0.8212 | 0.8238 | 0.8264 | 0.8289 | 0.8315 | 0.8340 | 0.8365 | 0.8389 |
| 1.00 | 0.8413 | 0.8438 | 0.8461 | 0.8485 | 0.8508 | 0.8531 | 0.8554 | 0.8577 | 0.8599 | 0.8621 |
| 1.10 | 0.8643 | 0.8665 | 0.8686 | 0.8708 | 0.8729 | 0.8749 | 0.8770 | 0.8790 | 0.8810 | 0.8830 |
| 1.20 | 0.8849 | 0.8869 | 0.8888 | 0.8907 | 0.8925 | 0.8944 | 0.8962 | 0.8980 | 0.8997 | 0.9015 |
| 1.30 | 0.9032 | 0.9049 | 0.9066 | 0.9082 | 0.9099 | 0.9115 | 0.9131 | 0.9147 | 0.9162 | 0.9177 |
| 1.40 | 0.9192 | 0.9207 | 0.9222 | 0.9236 | 0.9251 | 0.9265 | 0.9279 | 0.9292 | 0.9306 | 0.9319 |
| 1.50 | 0.9332 | 0.9345 | 0.9357 | 0.9370 | 0.9382 | 0.9394 | 0.9406 | 0.9418 | 0.9429 | 0.9441 |
| 1.60 | 0.9452 | 0.9463 | 0.9474 | 0.9484 | 0.9495 | 0.9505 | 0.9515 | 0.9525 | 0.9535 | 0.9545 |
| 1.70 | 0.9554 | 0.9564 | 0.9573 | 0.9582 | 0.9591 | 0.9599 | 0.9608 | 0.9616 | 0.9625 | 0.9633 |
| 1.80 | 0.9641 | 0.9649 | 0.9656 | 0.9664 | 0.9671 | 0.9678 | 0.9686 | 0.9693 | 0.9699 | 0.9706 |
| 1.90 | 0.9713 | 0.9719 | 0.9726 | 0.9732 | 0.9738 | 0.9744 | 0.9750 | 0.9756 | 0.9761 | 0.9767 |
| 2.00 | 0.9772 | 0.9778 | 0.9783 | 0.9788 | 0.9793 | 0.9798 | 0.9803 | 0.9808 | 0.9812 | 0.9817 |
| 2.10 | 0.9821 | 0.9826 | 0.9830 | 0.9834 | 0.9838 | 0.9842 | 0.9846 | 0.985  | 0.9854 | 0.9857 |
| 2.20 | 0.9861 | 0.9864 | 0.9868 | 0.9871 | 0.9875 | 0.9878 | 0.9881 | 0.9884 | 0.9887 | 0.9890 |
| 2.30 | 0.9893 | 0.9896 | 0.9898 | 0.9901 | 0.9904 | 0.9906 | 0.9909 | 0.9911 | 0.9913 | 0.9916 |
| 2.40 | 0.9918 | 0.9920 | 0.9922 | 0.9925 | 0.9927 | 0.9929 | 0.9931 | 0.9932 | 0.9934 | 0.9936 |
| 2.50 | 0.9938 | 0.9940 | 0.9941 | 0.9943 | 0.9945 | 0.9946 | 0.9948 | 0.9949 | 0.9951 | 0.9952 |





| 2.60 | 0.9953 | 0.9955 | 0.9956 | 0.9957 | 0.9959 | 0.9960 | 0.9961 | 0.9962 | 0.9963 | 0.9964 |
| 2.70 | 0.9965 | 0.9966 | 0.9967 | 0.9968 | 0.9969 | 0.9970 | 0.9971 | 0.9972 | 0.9973 | 0.9974 |
| 2.80 | 0.9974 | 0.9975 | 0.9976 | 0.9977 | 0.9977 | 0.9978 | 0.9979 | 0.9979 | 0.9980 | 0.9981 |
| 2.90 | 0.9981 | 0.9982 | 0.9982 | 0.9983 | 0.9984 | 0.9984 | 0.9985 | 0.9985 | 0.9986 | 0.9986 |
| 3.00 | 0.9987 | 0.9987 | 0.9987 | 0.9988 | 0.9988 | 0.9989 | 0.9989 | 0.9989 | 0.9990 | 0.9990 |
| 3.10 | 0.9990 | 0.9991 | 0.9991 | 0.9991 | 0.9992 | 0.9992 | 0.9992 | 0.9992 | 0.9993 | 0.9993 |
| 3.20 | 0.9993 | 0.9993 | 0.9994 | 0.9994 | 0.9994 | 0.9994 | 0.9994 | 0.9995 | 0.9995 | 0.9995 |
| 3.30 | 0.9995 | 0.9995 | 0.9995 | 0.9996 | 0.9996 | 0.9996 | 0.9996 | 0.9996 | 0.9996 | 0.9997 |
| 3.40 | 0.9997 | 0.9997 | 0.9997 | 0.9997 | 0.9997 | 0.9997 | 0.9997 | 0.9997 | 0.9997 | 0.9998 |

**Data availability**

All raw data can be provided by the corresponding authors upon request. The vertical column of ice water content (IWC) and mesospheric temperature data are from the SOFIE portal (http://sofie.gats-inc.com/). Additional temperature data are from the MLS website (https://aura.gsfc.nasa.gov/mls.html). The solar index Y10 is supported by the materials from "The solar and geomagnetic inputs into the JB2008 thermospheric density model for use by CIRA08 and ISO 14222, 2010".

**Acknowledgments**

This work is supported by the National Natural Science Foundation of China (NO. 41974178). We acknowledge the data usage from SOFIE onboard AIM, and MLS onboard Aura. The first author would like to thank Prof. Jia Yue from Catholic University of America for his kind suggestions on this work.

**Author information**

Affiliations

Department of Geophysics, College of the Geology Engineering and Geomatics, Chang'an University, Xi'an, 710054, China

Shican Qiu, Mengzhen Yuan & Zhanming Zhang

Key Laboratory of Geospace Environment, Chinese Academy of Sciences, University of Science & Technology of China, Hefei, Anhui, 230026

Mengzhen Yuan, Zhanming Zhang, Chengyun Yang & Xiankang Dou

Center for Environmental Research and Earth Sciences (CERES), Salem, MA 01970, USA

Willie Soon

Institute of Earth Physics and Space Science (ELKH EPSS), 9400, Sopron, Hungary





Willie Soon

Instituto De Geofísica, Universidad Nacional Autónoma De México, Mexico City, Mexico

        Victor Manuel Velasco Herrera

**Author contributions**

Shican Qiu conceived this study and wrote this manuscript.

Mengzhen Yuan performed data analysis.

Willie Soon was in charge of the organization and English polishing of the whole manuscript.

Mingjiao Jia prepared Fig. 2 and gave some useful comments on the content.

Victor Manuel Velasco Herrera did data analysis on correlations.

Chengyun Yang discussed the data results.

Xiankang Dou conceived this study and supported the first author for her research work.

**Competing interests**

The authors declare that they have no conflict of interest.

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





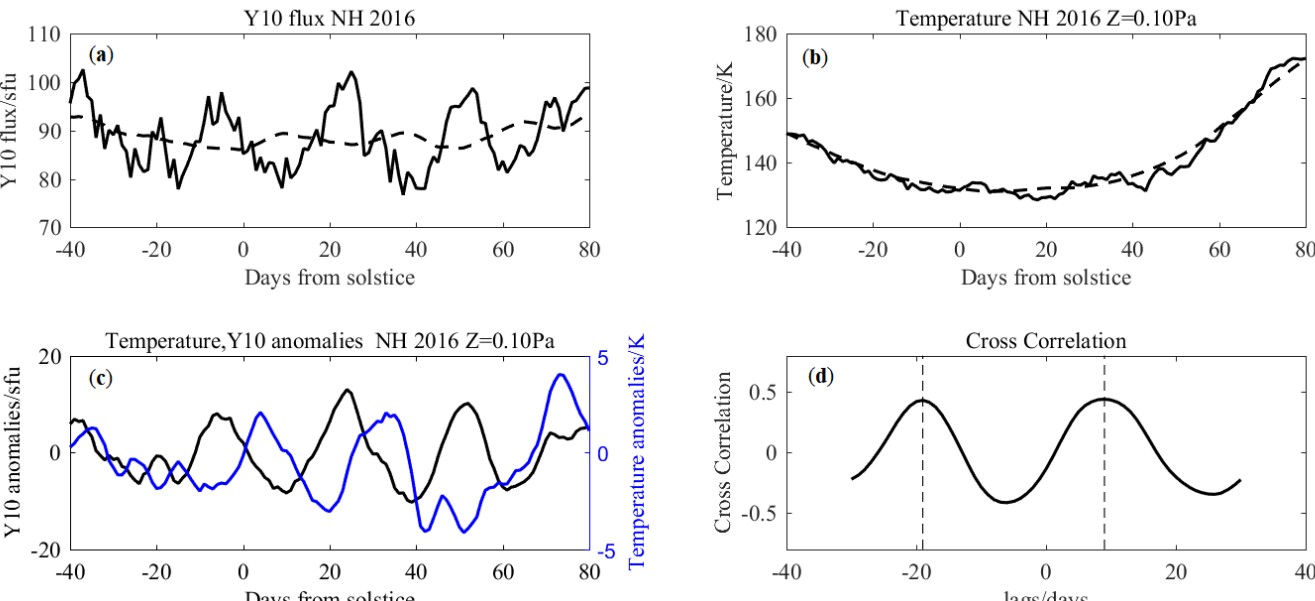

**Figure 1.** (a) The solar radiation index Y10 (solid line) and its 35-day running mean (dotted line) during the NH PMC season in 2016. (b) The average temperature (solid line) and its 35-day running mean (dotted line) in the latitude range of 60°–80° at 0.10 pa atmospheric pressure during the 2016 NH PMC season. (c) Y10 anomaly (black line) and temperature anomaly (blue line) to relative summer solstice days. (d) The correlation coefficient between Y10 anomaly and temperature anomaly at different time lag.



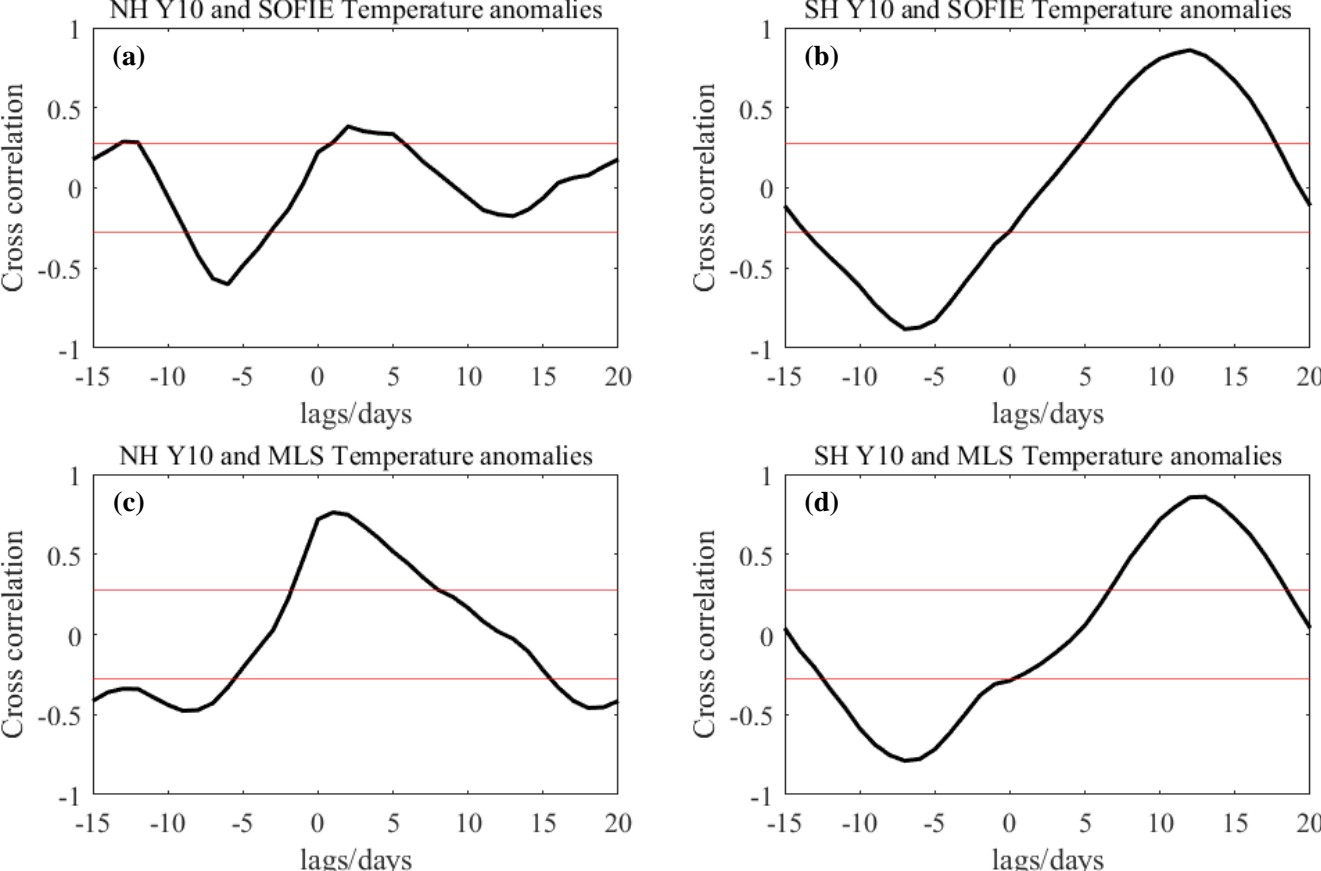

**Figure 2.** (a) The correlation curve of temperature anomaly from SOFIF and Y10 anomaly in the NH calculated through SEA. (b) The correlation curve of SOFIF temperature anomaly and Y10 anomaly in the SH. (c) The correlation curve of temperature anomaly from MLS and Y10 anomaly in the NH. (d) The correlation curve of MLS temperature anomaly and Y10 anomaly in the SH. The red line is the correlation coefficient corresponding to statistical significance equal to 95%.





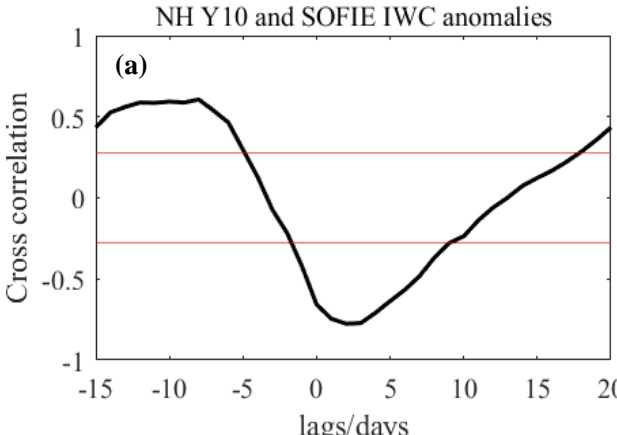 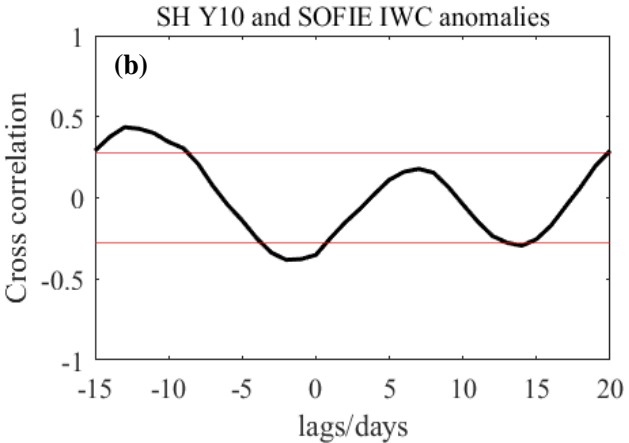


**Figure 3.** (a) The correlation curve between the IWC anomaly and Y10 anomaly in the NH calculated by SEA. (b) The correlation curve in the SH. The red line is the correlation coefficient corresponding to the statistical significance equal to 95%.





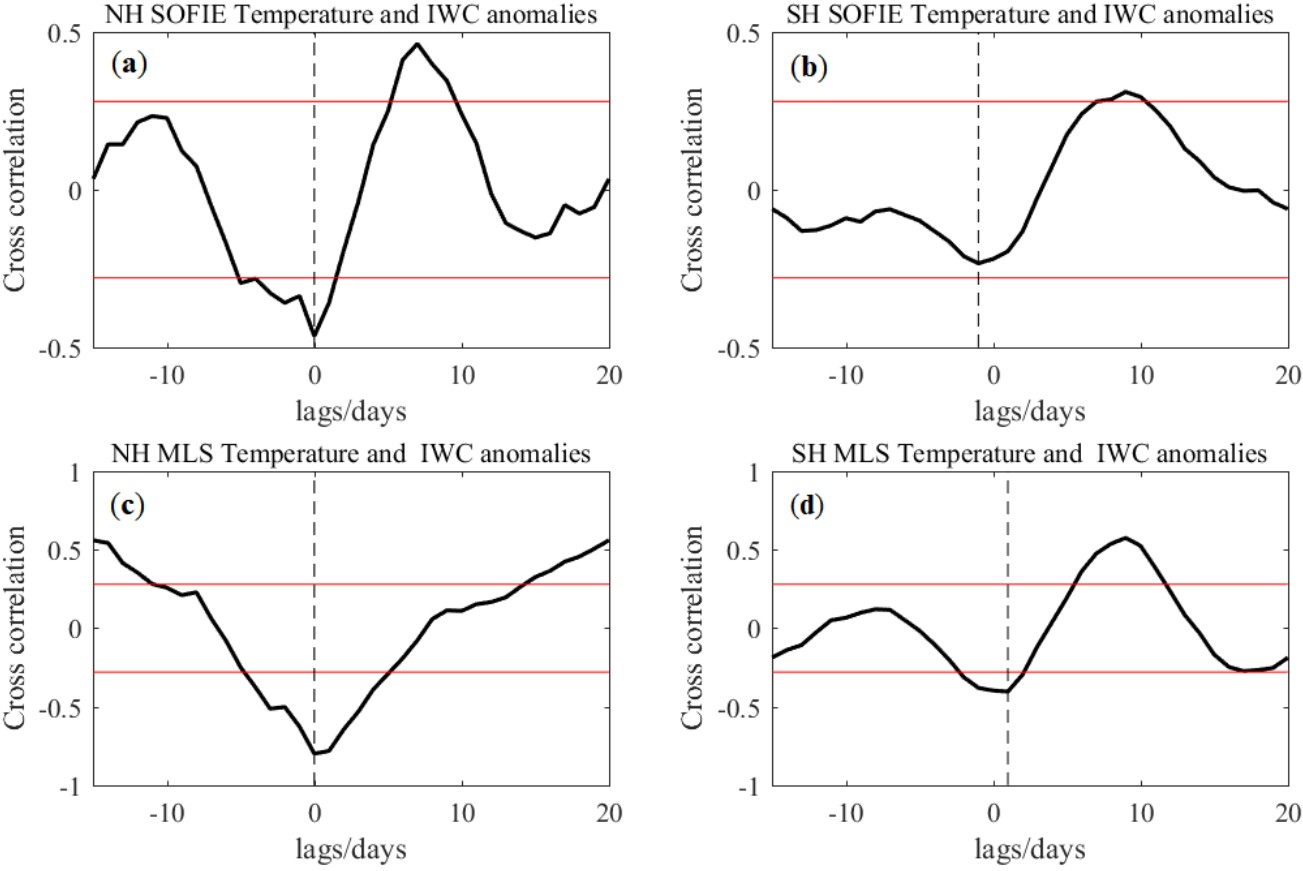

**Figure 4.** (a) The correlation curve of SOFIF temperature anomaly and IWC anomaly in the NH calculated by SEA. (b) The correlation curve of SOFIF temperature anomaly and IWC anomaly in the SH. (c) The correlation curve between MLS temperature anomaly and IWC anomaly in the NH. (d) The correlation curve between MLS temperature anomaly and IWC anomaly in the SH. The red line points out the correlation coefficient of statistical significance equal to 95 %.





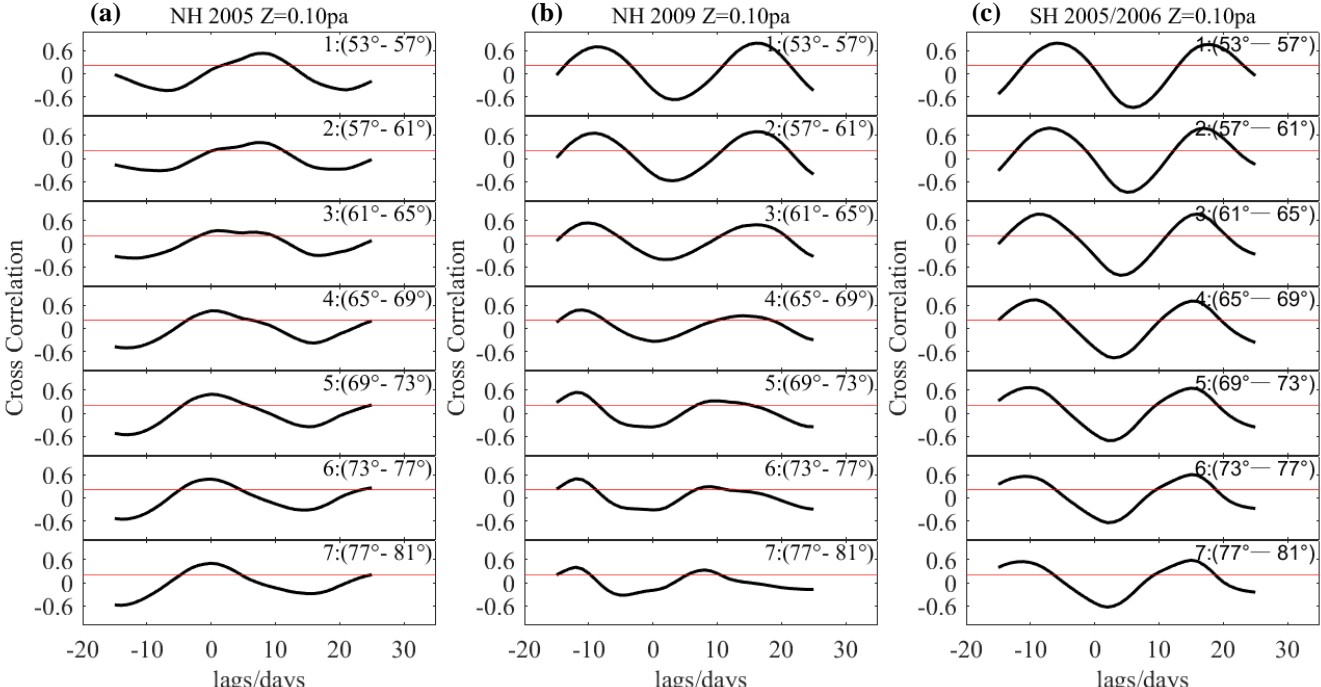


**Figure 5.** The correlation curves between temperature anomalies and Y10 anomalies for the PMC seasons of NH 2005 (a), NH 2009 (b) and SH 2005/2006 (c) at 0.10pa atmospheric pressure for different latitude ranges. The temperature is averaged in the corresponding latitude range, and the red line is the correlation coefficient of statistical significance equal to 95 %.

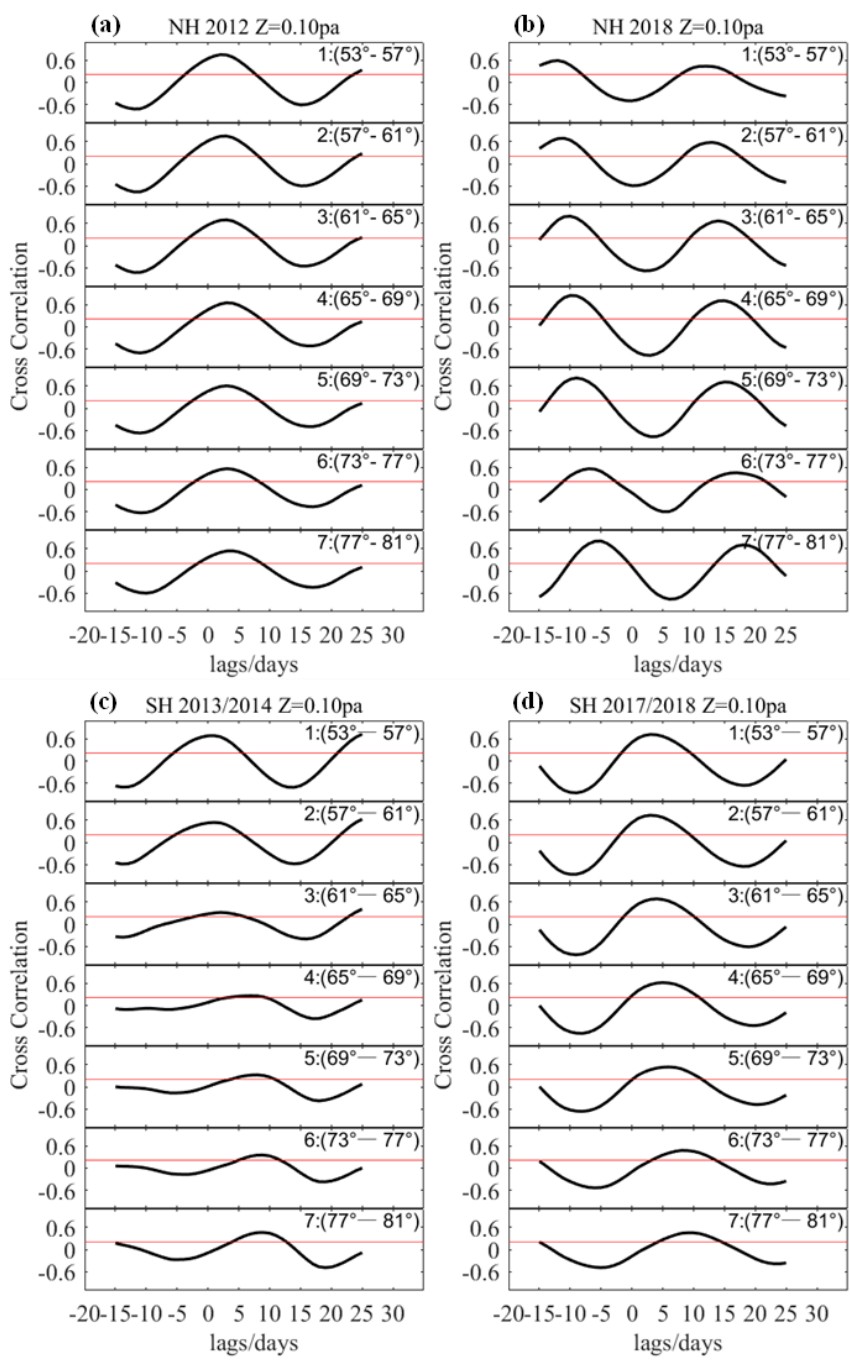

**Figure 6.** The correlation curves between temperature anomalies and Y10 anomalies for the PMC seasons of NH 2012 (a), NH 2018 (b), SH 2013/2014 (c) and SH 2017/2018 (d) at 0.10pa atmospheric pressure for different latitude ranges. The temperature is averaged in the corresponding latitude range, and the red line is the correlation coefficient of statistical significance equal to 95 %.





**Figure 7.** The correlation curves between temperature anomalies and Y10 anomalies for the PMC seasons of NH 2016 (a), NH 2019 (b), SH 2008/2009 (c), SH 2012/2013 (d), SH 2014/2015 (e), and SH 2015/2016 (f) at 0.10pa atmospheric pressure for different latitude ranges. The temperature is averaged in the corresponding latitude range, and the red line is the correlation coefficient of statistical significance equal to 95 %.





**Table 1.** The key dates of the selected PMC seasons used for the SEA

| PMC Season | Key Date (Day from Solstice) | PMC Season | Key Date (Day from Solstice) |
|---|---|---|---|
| NH 2008 | −5 | SH 2007/2008 | 13 |
| NH 2009 | 14 | SH 2008/2009 | 26 |
| NH 2010 | * | SH 2009/2010 | −4 |
| NH 2011 | 0 | SH 2010/2011 | −10 |
| NH 2012 | −10 | SH 2011/2012 | * |
| NH 2013 | * | SH 2012/2013 | −5 |
| NH 2014 | 17 | SH 2013/2014 | 15 |
| NH 2015 | 17 | SH 2014/2015 | 18 |

\* Represents no key date for the PMC seasons



**Table 2.** The average time lag (in days) of temperature response to the 27-day solar rotational modulation in the latitude range 53°–81° at different atmospheric pressure

| PMC Season | 0.1pa | 0.22pa | 0.46 |
|---|---|---|---|
| **NH 2009** | **12** | **\*** | **24** |
| NH 2012 | 3 | 2 | 2 |
| **NH 2016** | **9** | **11** | **13** |
| **NH 2019** | **14** | **\*** | **23** |
| **SH 2005/2006** | **16** | **17** | **19** |
| **SH 2008/2009** | **8** | **10** | **12** |
| SH 2012/2013 | 22 | 23 | 4 |
| **SH 2013/2014** | **5** | **7** | **9** |
| **SH 2014/2015** | **5** | **6** | **7** |
| **SH 2015/2016** | **14** | **17** | **18** |
| **SH 2017/2018** | **5** | **11** | **14** |
| SH 2018/2019 | \* | 14 | 13 |

\* Represents no key date for the PMC seasons