# Peer review of "The solar induced 27-day modulation on polar mesospheric cloud (PMC), based on the combined observations from SOFIE and MLS"

_Annales Geophysicae, 2022_

## Author Comment (AC1)

We would like to extend our cordial thanks to the reviewer for the valuable comments and constructive suggestions. We went through all of the comments thoroughly and revised the manuscript accordingly. All changes in red fonts have been marked in the revised manuscript. The explicit answers to the comments are given below in blue fonts.

Reviewer #1:

General Comments

This paper investigates how the 27-day solar rotation impacts polar mesospheric clouds (PMC), using observations from the SOFIE and MLS satellite instruments.

The paper reproduces previous investigations by several different authors, who used SOFIE observations to study how the 27-day solar rotation affects PMCs, temperature, and water vapor. The Authors need to better motivate their work, especially since they are repeating previous studies. Specifically, they should indicate what advancements they are after, and/or if there are issues with the previous studies that they hope to resolve.

Thanks for the comments. The authors would like to thank the valuable comments and suggestions in order to help us improve our manuscript. In this paper, we have studied the impact of solar radiation activity on the polar mesospheric clouds (PMCs), based on the correlations between the solar composite index Y10, the mesospheric temperature, and the ice water content (IWC). It is surprising but it is the first time that Y10 index is adopted as the solar parameter to study the responses of PMCs to the solar activities. More importantly. We have also found some new results from these observations.

The differences of temperature responses to Y10 between 16 SH and 16 NH PMC seasons are studied, based on the on the data measured by MLS at 0.46 pa (about 84 km altitude). The analyzed PMC seasons can be classified into three categories: 1) The cases have distinct 27-day solar cycle characteristics, and the time lag days increase with latitude (e.g., NH 2009, NH 2012, NH 2016, NH 2017, SH 2013 / 2014, SH 2015 / 2016). 2) The seasons have obvious 27-day period, but the time lags do not change with latitude (e.g., NH 2008, SH 2005 / 2006, SH 2008 / 2009, SH2013 / 2014, SH2014 / 2015, SH2017 / 2018). 3) The seasons have no obvious periodic variation. Our results indicate the atmospheric dynamics and the 27-day oscillations are possible reasons for the interannual differences in the mesospheric temperature response to Y10. Meanwhile, it is also found that the temperature responses exhibit similar characters every 10 years, such as NH 2005 vs. NH 2016, and SH 2004 / 2005 vs. SH 2014 / 2015. This result may be related to the 11-year cycle of the Sun.

We have modified our abstract and conclusion in the revised manuscript in order to summarize and explain our results more clearly.

Implementation of the English language could be greatly improved. I cite a few instances below, but certainly not all of them. This makes the paper difficult to read, and could certainly be used as grounds for rejection. Rather, I ask the Authors to revise the paper, and to possibly seek assistance with the writing before sending it back for review. The scientific community needs to uphold a high standard of writing and grammar, and this paper does not meet such a standard.

Thanks for the comment. We apologize about our confusing explanations very much. We have now revised the statements that were not clear and sentences that were too long throughout the revised manuscript. We have also revised the English wording and grammatical structures of the article.

Specific Comments

Line 12: It is certainly not true that temperature is considered by all to be the controlling factor. Some studies indicate that water vapor is the major factor determining PMC variability [e.g., Lubken et al., 2021]. You should make a fair statement here.

Thanks for the comment. We have highlighted the effect of water vapor on PMC around the line 12, and added this useful new reference in line 36.

Lines12-17: This sentence is too long.

Thanks for the comment. The sentence has been modified around the lines 12-17.

Line 30: What do you mean by "an increment of water"?

Thanks for the comment. It is "an increase of water vapor content". The sentence has been modified in the line 31.

Line 37: Eliminate "the" before PMC and mesospheric.

Thanks for the comment. This "the" before PMC and mesospheric has been eliminated in line 38.

Line 40-43: This sentence is too long.

Thanks for the comment. This sentence has been modified around the lines 41-44.

Line 72: SOFIE and AIM were defined above.

Thanks for the comment. The sentence has been modified in line 74.

Line 82: 1.5 degrees in what context? Is this the FOV width? Longitude?

Thanks for the comment. The 1.5 degrees is the latitude interval of sampling point in line 83.

Line 84: The MLS vertical resolution is roughly a scale height (~8 km) in the upper mesosphere. This makes it somewhat difficult for use in PMC studies, because PMCs have much finer vertical structure.

Thanks for the comment. The MLS scans vertically with spatial coverage of nearly global scale (with latitude of –82° to +82°). Thus, the MLS has a good latitude coverage. We need temperature and IWC data at different latitudes around 80 to 85 km. Your important observations concerning the vertical resolution of the MLS dataset and the fine details of the vertical structure of PMCs are beyond the scope and limitation of our current narrow study.

Line 85: Correlation between what parameters? What seasonal components are referred to? T & $H_2O$ I assume?

Thanks for the comment. We have rewritten this sentence in lines 87-89.

Line 88: A 35-day window will not remove the seasonal variation in polar summer. The seasonal variation in temperature for example, is more on the order of 100 days (Figure 1). 35 days rather removes natural variability and noise. Please clarify this discussion.

Thanks for the comment. We have revised the sentence in lines 89-90. A 35-day window is adopted to minimize random variability and noise. Then the time series of the anomalies caused by solar radiation can be obtained by subtracting the running mean of 35 days from the original time series. We have re-written the text in the revised manuscript.

Figure 1b: You do not state where the temperatures come from.

Thanks for the comment. We have added the source of temperatures in line 97.

Line 100: Add a reference here.

Thanks for the comment. We have added corresponding references in line 103.

Line 103: "serial numbers"? Perhaps you mean "sequential"

Thanks for the comment. We have corrected "serial numbers" to "sequence" in line 106.

Line 110-112:   The nomenclature T_anomalies is awkward, consider using Ta or DT (same comment for Y10).

Thanks for the comment. We have corrected "T_anomalies" to "DT" in lines 110-114 (same correction for Y10).

Lines 115-120: This discussion is hard to follow, and should be rewritten.

Thanks for the comment. We have rewritten the discussion in lines 116-120.

Line 156: Temperature is spelled wrong. Is it true that T drives the 27-day response in PMCs? What about water vapor? SOFIE measures $H_2O$, and you should look at that here as well. Perhaps there is a relationship that others have missed?

Thanks for the comment. We have respelled the word 'temperature'. The temperature and water vapor are two fundamental parameters that affect PMC. The temperature provides the conditions for crystallization, and water vapor provides the conditions for ice growth, both of which affect PMC in different . From the Eq. (6), we can see that S is exponentially related to temperature but linearly related to water vapor. The summer mesopause temperature is often lower than 140 K according to the observed data. Therefore, in order to simplify the relationship between PMCs and solar activity, the effect of water vapor can be ignored and only the temperature is chosen as the intermediate variable. In this paper we focus on how solar radiation affects PMCs by modulating the mesospheric temperatures.

     We have modified the discussion on the precondition for ignoring water vapor around lines 164 to 165.

Equation 6: It would be more appropriate to write S as the ratio of the $H_2O$ partial pressure over the saturation vapor pressure. You should then state that the denominator in Eqn. 6 is the saturation vapor pressure according to Marti and Mauersberger [1993].

Thanks for the comment. We have rewritten S as the ratio of the $H_2O$ partial pressure over the saturation vapor pressure. And we have state that the denominator in Eqn. 6 is the saturation vapor pressure.

Line 161: The accepted unit for pressure is hPa.

Thanks for the comment. We have corrected the unit for pressure in line 161.

Line 163: This does not mean that you can ignore water vapor. There are numerous papers that document a strong dependence of PMCs on water vapor, and you should certainly consider it as important here.

Thanks for the comment. The temperature and water vapor are two fundamental parameters that affect PMC. The temperature provides the conditions for crystallization, and water vapor provides the conditions for ice growth, both of which affect PMC in different forms. From the Eq. (6), we can see that S is exponentially related to temperature but linearly related to water vapor. The summer mesopause temperature is often lower than 140 K according to the observed data. Therefore, in order to simplify the relationship between PMCs and solar activity, the effect of water vapor can be ignored and only the temperature is chosen as the intermediate variable. In this paper we focus on how solar radiation affects PMCs by modulating the mesospheric temperature. But the role of the water vapor on the PMCs is beyond the scope of our current study.

Line 164: Are you looking at T from MLS or SOFIE or both? The pressure levels you select are below typical PMC heights (0.05 hPa or ~84 km). Is there a reason for this?

Thanks for the comment. This temperature data is observed by MLS only. We have added the source of T in the line 167. We have rechecked that the atmospheric pressure corresponding to typical PMC heights (~84km) is about 0.46 Pa.

Line 165 (and elsewhere): I think you mean hPa, not pa, for the unit of pressure.

Thanks for the comment. We have rechecked that the atmospheric pressure corresponding to typical PMC heights (~84 km) is about 0.46 Pa. The atmospheric pressure heights we studied here, 0.1 Pa, 0.22 Pa, and 0.46 Pa, are located at mesopause region between 80 and 90 km.

Lines 171-177: This discussion is poorly written and hard to understand.

Thanks for the comment. We have rewritten the discussion in lines 175-178.

Figure 2: "SOFIF" -> "SOFIE"

Thanks for the comment. We have corrected "SOFIF" to "SOFIE" in Figure 2.

Line 184: visibility -? variability?

Thanks for the comment. We have corrected "visibility" to "significance" in line 185.

Line 187: "exits" -> exists

Thanks for the comment. We have corrected "exits" to "exists" in line 188.

Line 192: I think you have confused the MLS and SOFIE results here. Furthermore, the anti-correlation is not significant for the SH using SOFIE T, and barely significant when using MLS T. This may indicate another pathway, perhaps water vapor.

Thanks for the comment. We have rechecked the data again to confirm that there are no confusion on MLS and SOFIE. From Figure 4, we can find that the statistical significance of the anti-correlation between temperature and IWC is indeed lower in the southern hemisphere than in the northern hemisphere. This could be due to water vapor or the asymmetrical dynamical behavior of the northern and southern hemispheres. But the role of the water vapor on the PMCs is beyond the scope of our current study.

Lines 203-205: What you offer as "an explanation" is simply a restatement of what is observed, and not an explanation of what may be causing it.

Thanks for the comment. We have modified the description around lines 205 to 206.

Figure 6 & Line 207: These results are puzzling. Panels a, c, and d show what is expected, with a positive correlation near zero time lag. Panel b seems to be completely opposite, with a negative correlation near zero time lag. This certainly warrants discussion, and perhaps some investigation to see if it is an error in the analysis, or something unique about that season.

Thanks for the comment. We have replaced Figure 5 to Figure 7 with new figures (now as Figures 5 and 6). We have rewritten this part around lines 200-225.

Figure 7: There are similar discrepancies here as were mentioned concerning Figure 6.

Thanks for the comment. We have replaced Figure 5 to Figure 7 with new figures (now as Figures 5 and 6). We have rewritten this part around lines 200-225.

Section 4.2: You show the correlation results vs. latitude in Figures 5-7 for 13 of the 16 PMC seasons examined. The presentation not in any logical order (i.e., by time…), and furthermore switches between NH and SH. I suggest that you rather show the SEA analysis vs. latitude in both hemispheres. Then, you can show one or two individual seasons of interest if they helps illustrate your ideas.

Thanks for the comment. We have resorted the new Figure 5 and Figure 6 by northern

and southern hemispheres by time.

Table 2: Are these results for SOFIE or MLS?

Thanks for the comment. These results are from MLS. We have added the source of results in line 228.

Line 228: Is it also possible that the change with height is a reflection of a dynamical process?  For example, vertical winds in the mesosphere are a few cm/s (or roughly 0.2 km/day), which is reminiscent of your change in time lag with height.

Thanks for the comment. We agree with the reviewer that this scenario is possible. However, for now, the effect of vertical wind on PMCs is still beyond our capability to resolve or answer. Therefore, we hope to study the effects of wind field and water vapor in the future work with other research team.

Section 5: The conclusions are concise, but are not clearly stated. This section needs to be rewritten.

Thanks for the comment. We have rewritten the conclusions in Section 5.

Thanks again for all the comments and questions. We have learned a lot from your patience and valuable suggestions. We hope our revised manuscript has been satisfactorily improved.

---

## Author Comment (AC2)

We would like to offer our sincere thanks to the reviewer for the valuable comments and constructive suggestions. We went through all of the comments thoroughly and revised the manuscript accordingly. All changes in red fonts have been marked in the revised manuscript. The explicit answers to the comments are given below in blue fonts.

Reviewer #2:

Review of "The solar induced 27-day modulation on polar mesospheric cloud (PMC), based on the combined observations from SOFIE and MLS" by Qiu et al.

The paper focuses on the investigation of 27-day oscillation of PMC based on satellite observations of IWC and temperature. The author conducted comprehensive data analysis and confirmed the correlation between the PMC variations with the solar activities, represented by the Y10 index. The presentation is reasonably clean the conclusions are solid. The data analysis algorithm follows Thurairajah et al., 2017 with more SOFIE data, coupled with MLS observations.

1. However, the role of MLS data are not clearly addressed in the paper. Shouldn't the simultaneous IWC and temperature measurements by SOFIE provide enough data already to establish their correlation? Involving a different instrument can cause unwanted bias in the results due to distinct sampling schemes, such as resolution and sensitivity issues, and may not help the investigation. If the author insists to include MLS data, the paper should articulate the motivation and, at least, provide some discussion on the consistency between the two instruments' measurements.

Thanks for the comment. We have highlighted the role of the data from MLS and modified the descriptions around lines 167 to 170. We use the temperature data from both MLS and SOFIE for the superposed epoch analysis (SEA), and the results between the two instruments are compared. The analyzed results for MLS and SOFIE are quite similar, indicating robustness of the results with minimal influence from how different instruments are used to acquire the data.

2. Note that the MLS IWC covers range starting from upper troposphere. Does the troposphere cloud ice affect the overall PMC results in the paper?

Thanks for the comment. The IWC data used in our manuscript is measured by SOFIE. According to Gordley et al., (2009) and Hervig et al., (2009a), it is shown clearly that the ice data from SOFIE can be retrieved at distinct altitude around the mesopause region.

Therefore, we think the troposphere cloud ice may have a very minimal impacts on the overall PMC results in the current study.

3. In addition, the objectives of this study are vaguely stated.

Thanks for the comment. We have rewritten the abstract and conclusions to revisit the purpose and results of this paper more clearly. In this paper we focus more narrowly (at least stated more clearly now than previous draft) on how solar radiation affects PMCs by modulating the mesospheric temperature.

4. The paper also lacks the discussions of the results. What do these new results help to understand the underline mechanism of the 27-day oscillation of PMC. The current version reads like an experimental report, simply repeating the analysis with more data and listing the outputs of the calculations. Overall, what are the new discoveries in this study and their contributions to the PMC dynamics and chemistry?

Thanks for the comment. In this paper, we have studied the impact of solar radiation activity on the polar mesospheric clouds (PMCs), based on the correlations between the composite solar index Y10, the mesospheric temperature, and the ice water content (IWC). It seems to be the first time that the composite Y10 index is adopted to study the responses of PMCs to solar activity variations. In addition, we wish to report on some new results obtained from these observations.

The differences of temperature responses to Y10 between 16 SH and 16 NH PMC seasons are studied, based on the on the data measured by MLS at 0.46 pa (about 84 km altitude). The analyzed PMC seasons can be classified into three categories: 1) The cases have distinct 27-day solar cycle characteristics, and the time lag days increase with latitude (e.g., NH 2009, NH 2012, NH 2016, NH 2017, SH 2013 / 2014, SH 2015 / 2016). 2) The seasons have obvious 27-day period, but the time lags do not change with latitude (e.g., NH 2008, SH 2005 / 2006, SH 2008 / 2009, SH2013 / 2014, SH2014 / 2015, SH2017 / 2018). 3) The seasons have no obvious periodic variation. Our results indicate the atmospheric dynamics and the 27-day oscillations are possible reasons for the interannual differences in the mesospheric temperature response to Y10. Meanwhile, it is also found that the temperature responses exhibit similar characters every 10 years, such as NH 2005 vs. NH 2016, and SH 2004 / 2005 vs. SH 2014 / 2015. This may be related to the 11-year cycle of the Sun. So, although somewhat preliminary, we think that it is good and meaningful to report evidence for the modulation of PMCs or PMCs-related properties by solar activity forcing on both the 27-day rotational and the 11-yr cycle timescales.

Minor comments:

Page 2, line43-45. This sentence reads somewhat strange. Please consider to rephrase.

Thanks for the comment. We have modified this sentence around lines 41 to 45.

Page 2, line 50 "…solar radiation will increase the mesopause temperature by heating". This statement is questionable. The MLT temperature variations are mostly

controlled by the dynamics and chemistry.

Thanks for the comment. We have revised that statement in line 50.

Page 3, line 60-62. This statement needs more clarification, I think. Is this a hypothesis based on observations or some model simulations?

Thanks for the comment. We have added the statement based on observations in line 62.

Are the temperatures in Figure 1 SOFIE temperature or MLS temperature?

Thanks for the comment. We have added the source of temperature in line 97. The temperature data used for Figure 1 is from MLS.

Page 6, line 162-164. This statement is only true when the temperature is low, I think. The equation is fine here, because the summer mesopause temperature is low. However, it would be good to set a limit.

Thanks for the comment. We have discussed a limit that when the temperature is low, in order to simplify the relationship between PMCs and solar activity, the effect of water vapor can be ignored around lines 164-165.

The MLS data site in the Data availability does not seem to demonstrate the data resource. At least it is not quite clear.

Thanks for the comment. We have changed the website of the data link in line 86.

Thanks again for all the comments and questions. We have learned a lot from your patience and valuable suggestions. We hope our revised manuscript has been satisfactorily improved.